# Vaccination induces rapid protection against bacterial pneumonia via training alveolar macrophage in mice

Hao Gu[1,2,3†], Xi Zeng[2,4†], Liusheng Peng[2], Chuanying Xiang[1], Yangyang Zhou[1], Xiaomin Zhang[1], Jixin Zhang[1], Ning Wang[1], Gang Guo[1], Yan Li[1], Kaiyun Liu[1], Jiang Gu[2], Hao Zeng[2], Yuan Zhuang[2], Haibo Li[2], Jinyong Zhang[2], Weijun Zhang[2], Quanming Zou[2]*, Yun Shi[1]*

[1]West China Biopharmaceutical Research Institute,West China Hospital, Sichuan University, Chengdu, China; [2]National Engineering Research Center of Immunological Products, Department of Microbiology and Biochemical Pharmacy, College of Pharmacy, Third Military Medical University, Chongqing, China; [3]Department of Clinical Laboratory, 971st Hospital of People's Liberation Army, Qingdao, China; [4]Department of Phamacy, The 78 th Group Army Hospital of Chinese People's Liberation Army, Mudanjiang, China

**\*For correspondence:**
qmzou2007@163.com (QZ);
shiyun@wchscu.cn (YS)

[†]These authors contributed equally to this work

**Competing interest:** The authors declare that no competing interests exist.

**Abstract** Vaccination strategies for rapid protection against multidrug-resistant bacterial infection are very important, especially for hospitalized patients who have high risk of exposure to these bacteria. However, few such vaccination strategies exist due to a shortage of knowledge supporting their rapid effect. Here, we demonstrated that a single intranasal immunization of inactivated whole cell of *Acinetobacter baumannii* elicits rapid protection against broad *A. baumannii*-infected pneumonia via training of innate immune response in *Rag1*$^{-/-}$ mice. Immunization-trained alveolar macrophages (AMs) showed enhanced TNF-α production upon restimulation. Adoptive transfer of immunization-trained AMs into naive mice mediated rapid protection against infection. Elevated TLR4 expression on vaccination-trained AMs contributed to rapid protection. Moreover, immunization-induced rapid protection was also seen in *Pseudomonas aeruginosa* and *Klebsiella pneumoniae* pneumonia models, but not in *Staphylococcus aureus* and *Streptococcus pneumoniae* model. Our data reveal that a single intranasal immunization induces rapid and efficient protection against certain Gram-negative bacterial pneumonia via training AMs response, which highlights the importance and the possibility of harnessing trained immunity of AMs to design rapid-effecting vaccine.

## Introduction

The multidrug-resistant (MDR) bacteria, including *Acinetobacter baumannii*, *Pseudomonas aeruginosa*, *Klebsiella pneumoniae*, *Escherichia coli*, and *Staphylococcus aureus,* pose a great threat to global public health (*Tacconelli et al., 2018*). Pneumonia caused by MDR bacteria is a major cause of morbidity and mortality, especially in hospitalized patients (*Gonzalez-Villoria and Valverde-Garduno, 2016*; *Micek et al., 2015*; *Zilberberg et al., 2016*). The continuing spread of antimicrobial resistance has made treating MDR bacterial pneumonia extremely difficult. Vaccination has been proposed as a promising strategy for controlling MDR bacterial infections (*Jansen et al., 2018*; *Rappuoli et al., 2017*; *Williams, 2007*). Current vaccination strategies usually require multiple injections weeks or months apart, which limit them to rapidly prevent infections for inpatients. However, hospitalized patients have an especially high risk for exposure to MDR bacteria (*Pachón and McConnell, 2014*).

Therefore, rapid efficacy induced by vaccination is vital for vaccine development against MDR bacteria (*Pachón and McConnell, 2014*).

Induction of immunological memory is the central goal of vaccination. Immunological memory protects against infections by enabling a quicker and stronger immune response to a previously encountered antigen (*Farber et al., 2016*). Classically, immune memory is thought to be exclusively mediated by adaptive T and B cell responses. These responses are highly specific to antigen, but take days or weeks to become effective. Another part of the immune system, innate immune response, provides an initial, relatively nonspecific response to infection within hours to days without immunological memory. However, in the past decade, evidence has emerged showing that innate immune cells such as monocytes, macrophage, and NK cells can also build long-term memory through epigenetic and metabolic reprogramming of cells. This memory termed 'trained immunity' or 'trained innate immunity" produces hyperresponsiveness upon restimulation in these cells (*Netea et al., 2016*; *Netea and Joosten, 2018*; *Netea et al., 2011*). The rapidity of innate immune response leads us to speculate that trained innate immunity might effectively serve as the underlying mechanism for vaccination-induced rapid protection.

Here, we demonstrated that a single intranasal immunization of inactivated whole cell (IWC) induced rapid, efficient, and broad protection against certain Gram-negative bacterial pneumonia, which was dependent on trained innate immunity mediated by alveolar macrophages (AMs). These findings highlight the possibility to harness the trained immunity of AMs to design a vaccine with rapid efficacy against pulmonary infection.

## Results

### Rapid protection against *A. baumannii* pneumonia by a single intranasal vaccination

To evaluate whether vaccination can elicit rapid protection, mice were immunized intranasally (i.n.) with an IWC of *A. baumannii* strain LAC-4 and infected intratracheally (i.t.) with a lethal dose of LAC-4 2 days or 7 days later (*Figure 1A*). The control mice succumbed to the infection, whereas all IWC-vaccinated mice survived when mice were challenged 2 days or 7 days post immunization (*Figure 1B*). Consistent with the survival rate, bacterial burdens in lungs and blood of vaccinated mice were significantly lower than those in the control group at 24 hr post infection (hpi; *Figure 1C*). Histopathology of lung tissues showed reduced lung damage and decreased inflammatory cells infiltration in IWC-immunized mice (*Figure 1D* and *Figure 1—figure supplement 1*). When challenging the mice at day 7 post immunization, the pro-inflammatory cytokines of IL-6 in lungs (*Figure 1E*) and serum levels of IL-6 and TNF-α (*Figure 1F*) in the IWC-vaccinated group were significantly lower than those in the control group at 24 hpi. Expression of chemokines *Cxcl1*, *Cxcl2*, *Cxcl5*, *Cxcl10*, and *Ccl2* was also significantly reduced in the lungs of vaccinated mice at 24 hpi (*Figure 1G*). Inflammatory cells in the lungs were detected by the flow cytometry and gating strategy is shown in *Figure 1—figure supplement 2*. The results showed that the number of neutrophils in lungs of vaccinated mice was significantly lower than that of control group at 24 hpi (*Figure 1H*). Collectively, these findings indicate a single intranasal immunization with IWC elicits rapid and complete protection against pulmonary *A. baumannii* infection.

### Broad protection against *A. baumannii* pneumonia by immunization of IWC

To further characterize whether a single-strain IWC could generate protection against heterologous strains, we immunized mice with IWC of *A. baumannii* strain LAC-4 and challenged mice with clinical isolates SJZ24 and SJZ26 7 days later to observe the survival of mice. The results showed that a single intranasal immunization with IWC of LAC-4 also elicited rapid protection against pulmonary infection of clinical isolates SJZ24 and SJZ26 (*Figure 2A*). Due to limited hypervirulent strains of *A. baumannii* available, we did cross-immunization with IWC of different *A. baumannii* isolates and challenged mice with LAC-4. These results showed that intranasal immunization of different strains of SJZ24, SJZ26, SJZ09, or ATCC17978 all induced rapid and effective protection against lethal LAC-4 infection (*Figure 2B*). SJZ24 is an extremely antibiotic-resistant clinical isolate (*Zeng et al., 2019*), LAC-4 is a MDR strain, and ATCC17978 is a reference strain which is an antibiotic-sensitive strain (*Harris et al.,*

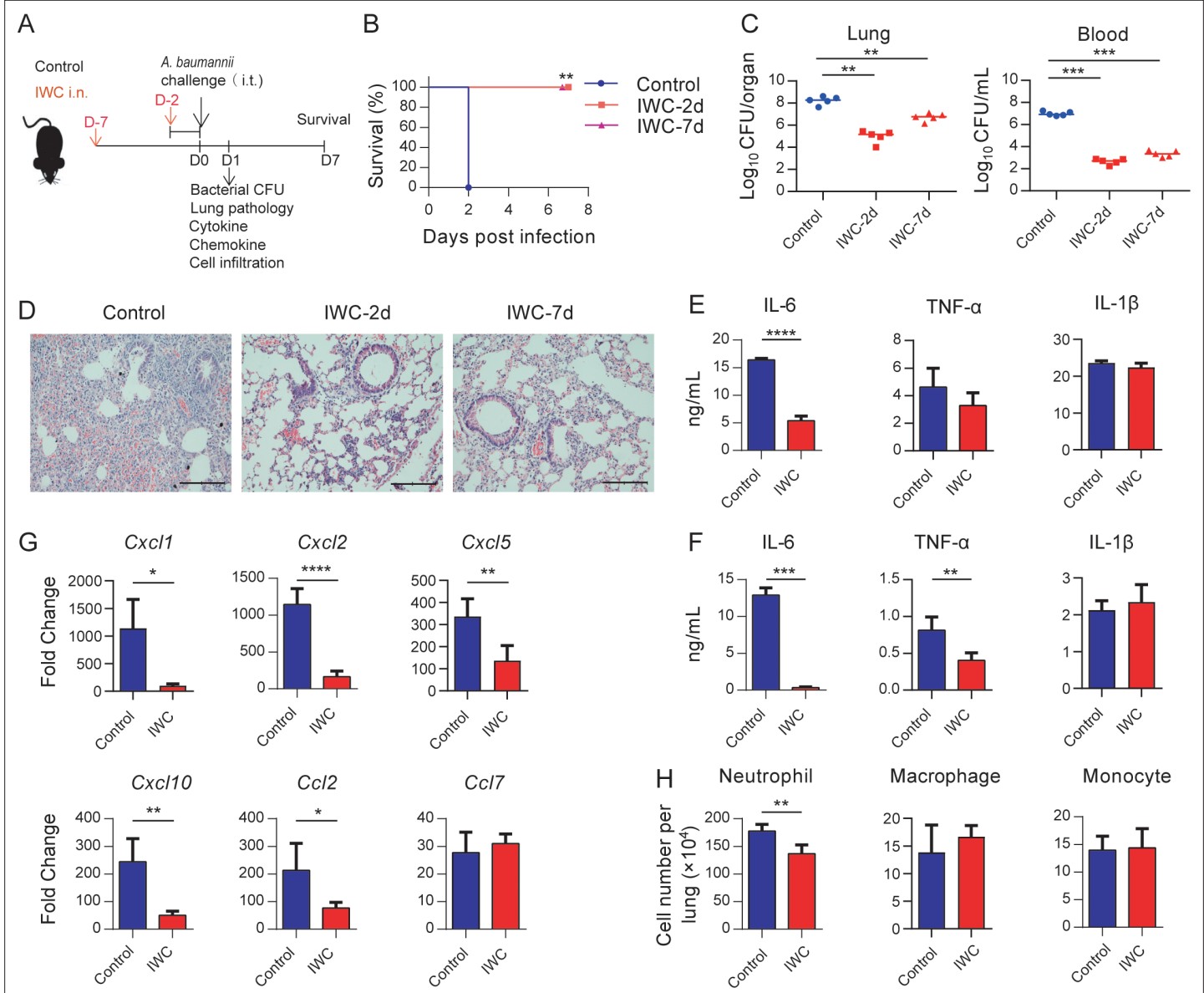

**Figure 1.** Rapid protection against *A. baumannii* pneumonia by a single intranasal vaccination. (**A**) Schematic diagram of the experimental procedure. C57BL/6 mice were immunized intranasally (i.n.) with inactivated whole cell (IWC) of *A. baumannii* strain LAC-4 and challenged intratracheally (i.t.) with LAC-4 at day 2 (IWC-2d) or day 7 (IWC-7d) after immunization (n = 5/group). (**B**) Survival of mice was recorded for 7 days. **p<0.01 determined by log-rank test. (**C**) Bacterial burdens in lungs and blood at 24 hr post infection (hpi) were determined. Each plot represents one mouse. The line indicates the median of the data. **p<0.01, ***p<0.001 evaluated by ordinary one-way ANOVA followed by Tukey`s multiple comparisons test. (**D**) Representative histopathological images of lungs at 24 hpi. Scale bars: 100 µm. (**E–G**) IWC-immunized mice were challenged at day 7 and were sacrificed at 24 hpi. Levels of inflammatory cytokines in the lungs (**E**) and serum (**F**) were detected by ELISA. (**G**) Transcriptional levels of chemokines in the lungs were detected by real-time PCR. (**H**) Numbers of neutrophils in the lungs were detected by flow cytometry. Data are mean ± SD. n = 4–5 mice/group. For (**E–H**), *p<0.05, **p<0.01, ***p<0.001, and ****p<0.0001, determined by two-tailed unpaired *t* test. Data are representative of at least two independent experiments.

The online version of this article includes the following source data and figure supplement(s) for figure 1:

**Source data 1.** Raw data for *Figure 1*.

**Figure supplement 1.** The histopathology images of lungs for each mouse in *Figure 1*.

**Figure supplement 2.** Gating strategy for detecting neutrophil, monocyte, and alveolar macrophages in lungs.

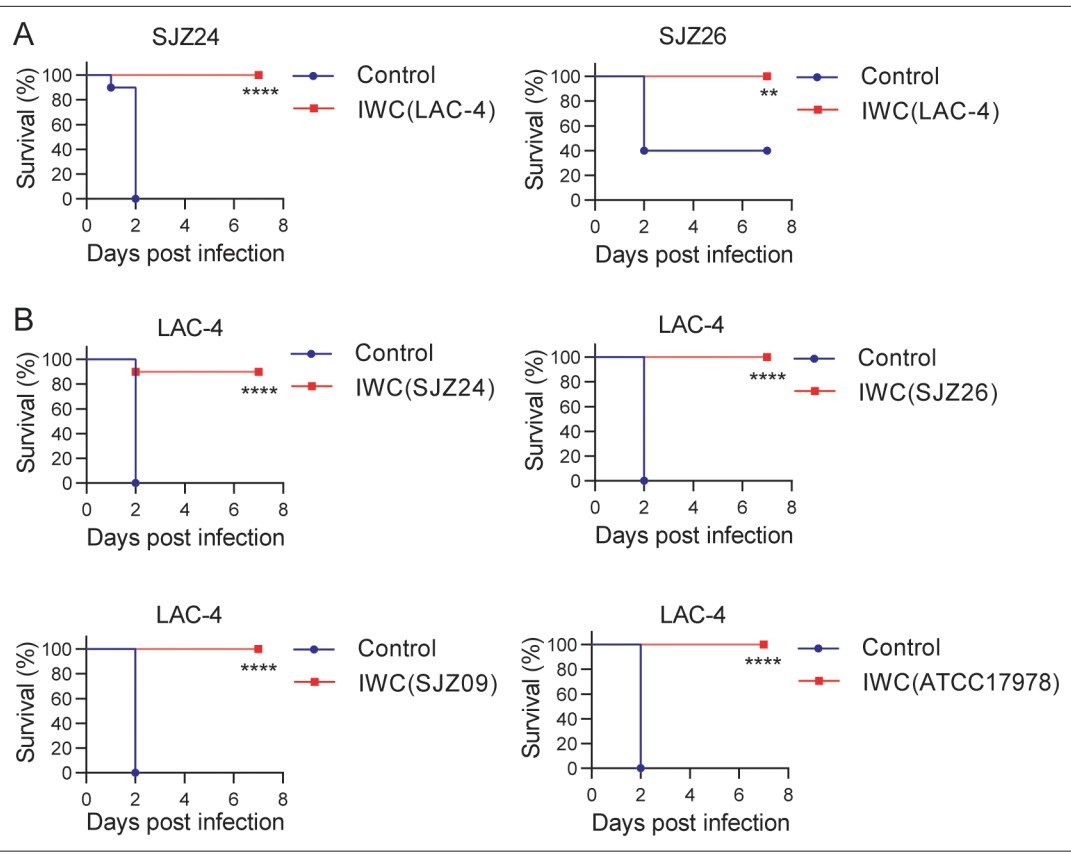

**Figure 2.** Broad protection against infection of clinical strains by intranasal immunization of inactivated whole cell (IWC) of *A. baumannii*. (**A**) Mice were immunized intranasally (i.n.) with IWC of LAC-4 on day 0. On day 7, mice were infected intratracheally (i.t.) with lethal dose of clinical strains of *A. baumannii* SJZ24 or SJZ26 (n = 10). (**B**) Mice were immunized i.n. with IWC of SJZ24, SJZ26, SJZ09, or ATCC17978 and challenged with lethal dose of LAC-4 7 days later. Survival rate of mice was monitored for 7 days after challenge. n = 10. **p<0.01, ****p<0.0001 determined by log-rank test. These experiments were repeated at least twice.

The online version of this article includes the following source data for figure 2:

**Source data 1.** Raw data for the survival rate.

*2013*). Together, these data suggested that intranasal immunization of IWC of *A. baumannii* can elicit broad protection against different clinical strains independent of drug resistance.

## Rapid immune memory induced by a single intranasal vaccination

Immunological memory is defined as functionally enhanced, quicker, and more effective response to pathogens that have been encountered previously, which is the basis of successful vaccines against subsequent infections (*Pollard and Bijker, 2021*). To assess whether the IWC-induced rapid protection is a result of immunological memory or a result of the persistent activation of innate immune responses, we measured the dynamic immune response after immunization of IWC from day 0 to day 7. In response to intranasal immunization of IWC, levels of TNF-α and IL-6 in lungs increased from day 1 to day 4 and completely declined to baseline by day 5 (*Figure 3A*), indicating that the host response rapidly primed and rest 5 days later. Then we challenged the mice with *A. baumannii* on day 7 after vaccination and assessed the cytokine levels in lungs early at 2 hpi. We found that TNF-α but not IL-6 and IL-1β levels was significantly higher in lungs from IWC-immunized mice than those from control mice (*Figure 3B*), suggesting that IWC-induced rapid TNF-α production might be responsible for the rapid protection. Further, mRNA levels of *Cxcl1*, *Cxcl2*, *Cxcl5*, *Cxcl10*, and *Ccl2* were higher in vaccinated mice than in control mice at early 2 hpi (*Figure 3C*). Meanwhile, consistent with the increased chemokine expression, vaccinated mice had significantly higher numbers of neutrophils and monocytes in their lungs than did control mice at 4 hpi (*Figure 3D*). These results indicate that IWC

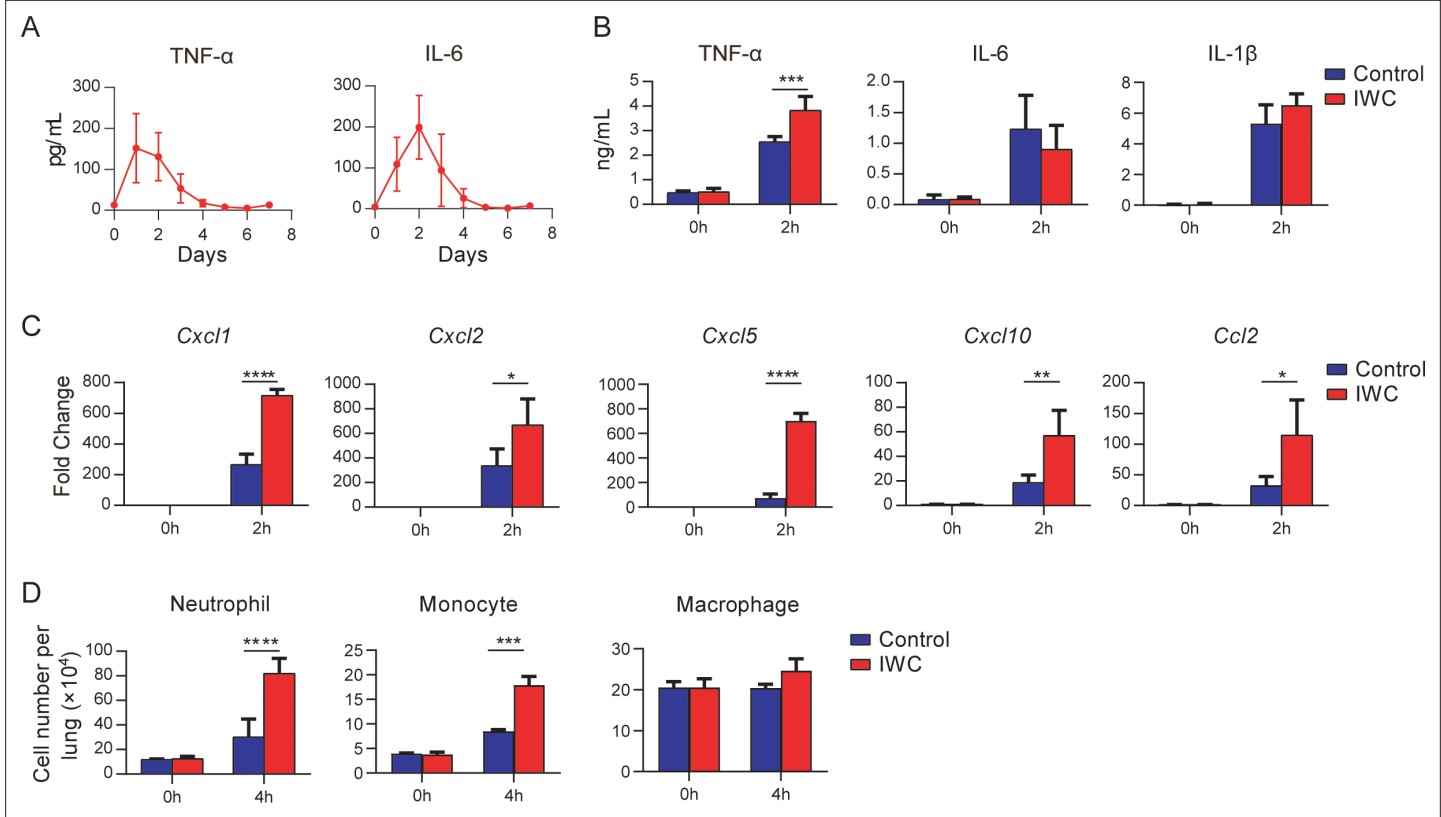

**Figure 3.** Rapid immune memory induced by a single intranasal vaccination. (**A**) Dynamic responses of TNF-α and IL-6 in the lungs of *A. baumannii* inactivated whole cell (IWC) (LAC-4)-immunized mice (n = 3 per timepoint). (**B–D**) IWC (LAC-4)-immunized mice were challenged intratracheally (i.t.) with LAC-4 at day 7 after immunization. (**B**) Levels of TNF-α, IL-6, and IL-1β at 0 hr and 2 hr post infection (hpi) in the lungs were measured by ELISA. (**C**) Transcriptional levels of chemokines in the lungs at 0 hr and 2 hpi were assessed by real-time PCR. (**D**) Numbers of neutrophils, monocytes, and macrophages in the lungs of mice were determined by flow cytometry. Data are presented as mean ± SD. (n = 3–4 mice/group). *p<0.05; **p<0.01, ***p<0.001, ****p<0.0001, ordinary two-way ANOVA. Data are representative of two independent experiments.

The online version of this article includes the following source data for figure 3:

**Source data 1.** Raw data for *Figure 3*.

immunization induces quick and enhanced responses upon *A. baumannii* challenge on day 7 after vaccination, which is an enhanced recall response of immune response.

## Vaccination-induced rapid protection is dependent on trained innate immunity

To further explore the potential mechanism of rapid effect of vaccination, *Rag1⁻ᐟ⁻* mice (which lack mature T and B cells) were immunized i.n. with IWC (LAC-4) and then challenged with a lethal dose of LAC-4 to determine which part of immune response is responsible for the rapid protection. Results showed that IWC provided rapid and effective protection both in WT and *Rag1⁻ᐟ⁻* mice at day 7 after immunization (*Figure 4A*). There is no significant difference between WT and *Rag1⁻ᐟ⁻* mice in terms of survival of the IWC-immunized group (*Figure 4A*, p=0.30, log-rank test). RNA sequencing analyses (RNA-seq) of lung tissue at day 7 after immunization and 24 hr after *A. baumannii* challenge revealed significant differentially expressed genes (DEGs) between transcriptional profiles of vaccinated *Rag1⁻ᐟ⁻* mice and those of control *Rag1⁻ᐟ⁻* mice (*Figure 4B* and *Figure 4—figure supplement 1A*). There were a total of 2401 DEGs between these two groups; 1084 upregulated genes and 1317 downregulated in the vaccinated mice (*Figure 4—figure supplement 1B*). Gene Ontology (GO) analysis of DEGs revealed that genes associated with inflammatory response, response to molecule of bacterial origin, and response to liposaccharide were significantly downregulated in vaccinated mice at 24 hpi (*Figure 4C* and *Figure 4—figure supplement 1C*). The expression of inflammation-related genes including *Il6*, *Cxcl2*, *Cxcl10*, and *Ccl2* was notably lower in vaccinated-*Rag1⁻ᐟ⁻* mice

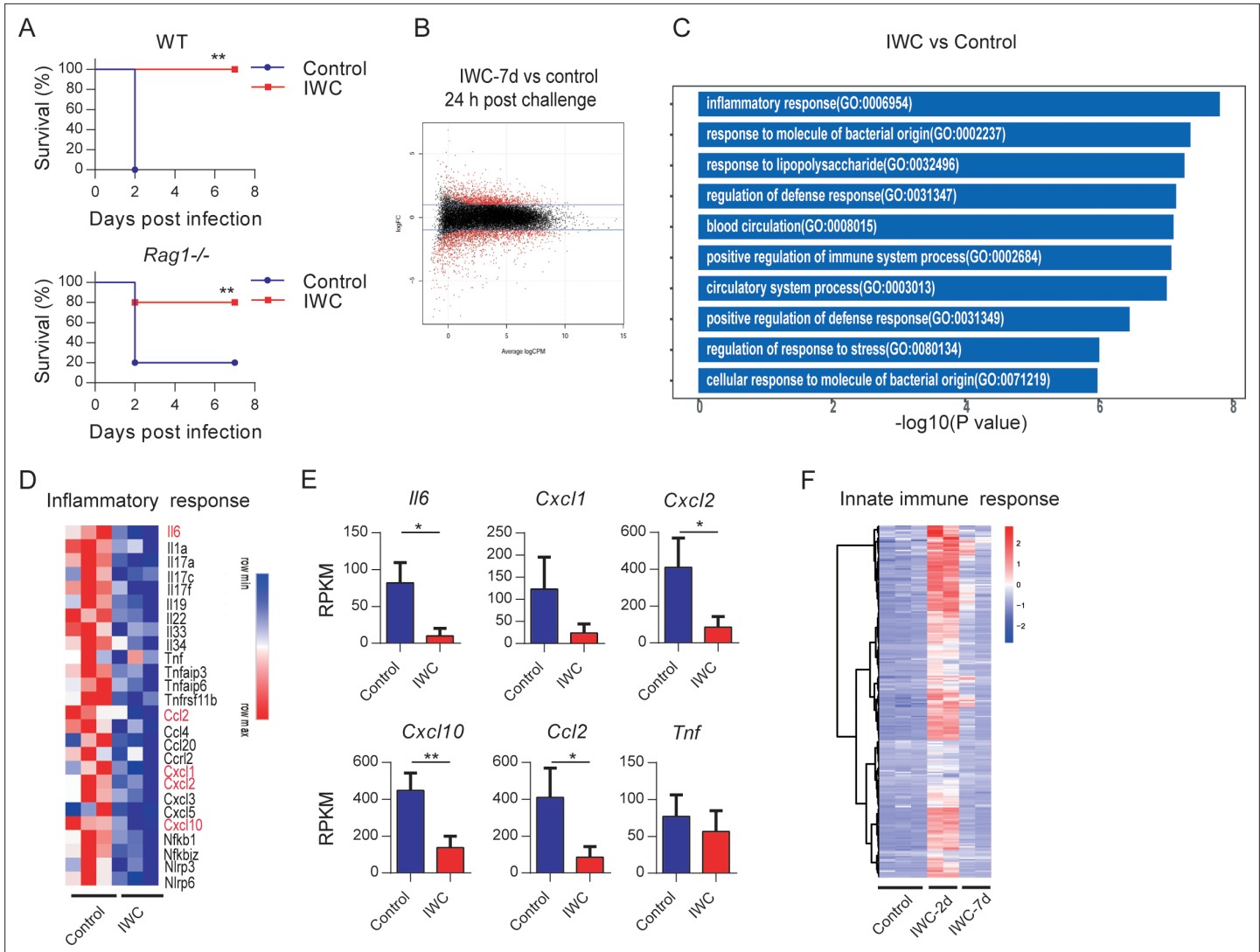

**Figure 4.** Trained innate immunity mediates vaccination-induced rapid protection. (**A**) Survival of WT and *Rag1⁻/⁻* mice immunized intranasally (i.n.) with inactivated whole cell (IWC) (LAC-4) or PBS, challenged intratracheally (i.t.) by lethal LAC-4 7 days later (n = 5 for WT mice, n = 10 for *Rag1⁻/⁻* mice). \*\*p<0.01 compared to control calculated by log-rank test. Data are representative of two independent experiments. (**B–E**) *Rag1⁻/⁻* mice were immunized with IWC (LAC-4) or PBS as control for 7 days and challenged with LAC-4. The lung tissue was collected to perform RNA-seq. (**B**) MA plot of the differentially expressed genes (DEGs) of IWC (LAC-4)-immunized lung (n = 3) vs. control lung of *Rag1⁻/⁻* mice (n = 3) at 24 hr post LAC-4 challenge. X-axis represents average counts per million (logCPM) and Y-axis represents log fold changes (logFC) in IWC-immunized mice vs. control mice. The blue line is the threshold, logFC >1 means upregulation and logFC < –1 means downregulation. (**C**) Top 10 Gene Ontology (GO) enrichment terms of downregulated DEGs in the IWC-immunized group at 24 hr post infection (hpi). (**D**) Heatmap of DEGs related to inflammatory response is shown. False discovery rate (FDR) < 0.05. (**E**) Reads per kilobase per million reads (RPKM) of inflammatory and chemokine genes. Data are mean ± SD. \*p<0.05, \*\*p<0.01 determined by two-tailed unpaired *t* test. (**F**) The heatmap of innate immune response-related genes (GO_0045087) of lung samples from control (n = 3), IWC (LAC-4)-immunized at day 2 (IWC-2d) (n = 2), and day 7 (IWC-7d) (n = 2) in *Rag1⁻/⁻* mice.

The online version of this article includes the following source data and figure supplement(s) for figure 4:

**Source data 1.** Raw data for *Figure 4A,E*.

**Source data 2.** RNA-seq results of reads per kilobase per million reads (RPKM) for all samples and all genes.

**Figure supplement 1.** Supplementary RNA-seq results.

than those in control mice at 24 hpi (*Figure 4D,E* and *Figure 4—figure supplement 1D*). These data indicate that IWC immunization induces rapid protection in *Rag1⁻/⁻* mice, highlighting the role of innate immune response in vaccination-induced rapid protection. We also analyzed the dynamic transcriptional response to IWC immunization in *Rag1⁻/⁻* mice and found that the innate immune response

was activated at day 2 and rested at day 7 (*Figure 4F*), which has a similar pattern to that in WT mice (*Figure 3*). Upon *A. baumannii* challenge at day 7 after immunization, the immunized mice exhibited different responses to those unimmunized mice (*Figure 4—figure supplement 1E*). These results indicate that IWC immunization induces a trained feature of innate immune response, which is critical for vaccination-induced rapid protection.

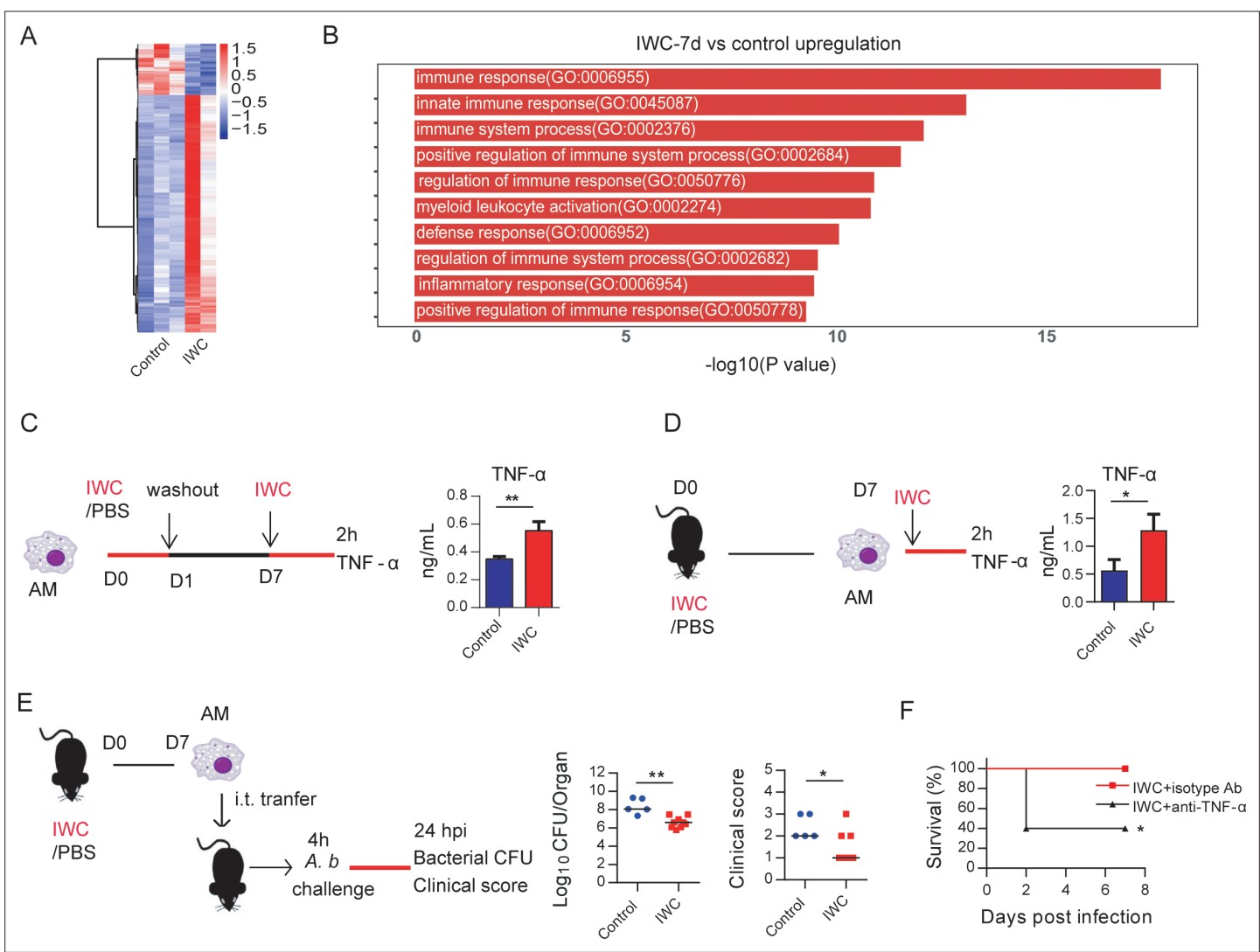

**Figure 5.** Trained immunity of alveolar macrophages (AMs) mediates rapid protection induced by vaccination. (**A**) Heatmap of differentially expressed genes (DEGs) in lungs of inactivated whole cell (IWC) (LAC-4)-immunized and control *Rag1*[-/-] mice at day 7 after immunization. (**B**) Top 10 Gene Ontology (GO) terms of upregulated DEGs in IWC-immunized *Rag1*[-/-] mice at day 7. (**C**) In vitro model of IWC (LAC-4)-trained AMs. (**D**) C57BL/6 mice were immunized with IWC (LAC-4), and recall responses of trained AMs to IWC (LAC-4) were evaluated ex vivo by detection of TNF-α production at 2 hr after stimulation. For (**C**) and (**D**), data are mean ± SD. n = 3. *p<0.05, **p<0.01, two-tailed unpaired *t* test. (**E**) Schema of evaluating roles of AMs in BALF of PBS or IWC (LAC-4)-immunized C57BL/6 mice. Bacterial burdens of lung and clinical scores at 24 hr post infection (hpi) were measured (n = 5–9). Each dot means a mice, and the line represents the median. *p<0.05, **p<0.01 determined by Mann–Whitney U test. (**F**) WT mice were immunized with IWC (LAC-4) for 7 days. Mice were treated intraperitoneally with anti-TNF-α antibody or isotype control then were challenged with lethal LAC-4 1 hr later. The survival of mice was monitored (n = 5). *p<0.05, calculated by log-rank test. Data are representative of two independent experiments.

The online version of this article includes the following source data and figure supplement(s) for figure 5:

**Source data 1.** Raw data for *Figure 5*.

**Figure supplement 1.** Transcriptional difference at day 7 after immunization of inactivated whole cell (IWC) (LAC-4) in *Rag1*[-/-] mice.

## Trained immunity of AMs mediates vaccination-induced rapid protection

Further, we analyzed the transcriptome change induced by vaccination at day 7 to identify DEGs associated with trained innate immunity. RNA-seq data showed a total of 308 DEGs in lungs of IWC-vaccinated *Rag1*[-/-] mice at day 7 (*Figure 5A* and *Figure 5—figure supplement 1A*). The upregulated 253 DEGs were enriched to myeloid leukocyte activation (*Figure 5B*), and these genes were enriched to macrophage-associated genes (*Figure 5—figure supplement 1B*). So, we reasoned that AMs, the predominant patrol myeloid cells in airways, might play a key role in vaccination-induced rapid protection. To test this hypothesis, we established a trained immunity model of AMs by stimulation with IWC in vitro. IWC-trained AMs induced an enhanced TNF-α production upon restimulation with IWC 7 days later (*Figure 5C*). This result indicates that IWC trains AMs directly. AMs from IWC-immunized or control mice at day 7 were sorted from bronchoalveolar lavage fluid (BALF) using CD11c[+] microbeads. Flow cytometry analysis with anti-CD11c and anti-F4/80 confirmed the purity of AMs was greater than 95% (*Figure 5—figure supplement 1C*). Sorted AMs were stimulated with IWC ex vivo for 2 hr. TNF-α production in vaccinated AMs after restimulation was significantly higher than that in control AMs (*Figure 5D*). Thus, AMs could be trained by IWC with functional reprogramming, showing increased TNF-α production to a previously encountered stimulus. Further, the purified AMs from *A. baumannii* IWC-immunized or control mice at day 7 were adoptively transferred into the airway of naïve mice by direct intratracheal instillation. Upon *A. baumannii* challenge, the lungs of mice that had received transfer of IWC-primed AMs had significantly lower bacterial burdens with alleviated clinical scores (*Figure 5E*). Further treatment of IWC-immunized mice with anti-TNF-α antibody before *A. baumannii* challenge resulted in reduced protection (*Figure 5F*). These results indicate that vaccination-trained AMs mediate rapid protection against infection via enhanced TNF-α production.

## Contribution of elevated TLR4 expression on trained AMs to rapid protection

RNA-seq reveals that genes related to myeloid leukocyte activation including TLRs were significantly upregulated on day 7 after *A. baumannii* IWC (LAC-4) immunization in *Rag1*[-/-] mice (*Figure 6A* and *Figure 6—figure supplement 1*). These results suggest that surface molecules associated with cell activation might be markers for trained AMs. Since TLR4 plays an important role in host recognition of Gram-negative bacteria, we hypothesized that IWC-trained AMs with elevated TLR4 expression might be more sensitive for second recall activation and enhanced function. RAN-seq data showed that lung TLR4 transcript significantly increased in response to IWC (LAC-4) immunization at days 2 and 7 (*Figure 6B*). The elevated TLR4 expression on BALF AMs at day 2 or day 7 after IWC (LAC-4) immunization was also confirmed by flow cytometry (*Figure 6C*). We also found that TLR4 expression on AMs was elevated at day 7 after IWC (LAC-4) training in vitro (*Figure 6D*). Further, we found that the rapid protective effect of IWC-vaccination was significantly reduced in *Tlr4*[-/-] mice than in WT mice (*Figure 6E*). Accordingly, IWC vaccination could not reduce bacterial burdens in lungs and blood in *Tlr4*[-/-] mice upon *A. baumannii* challenge (*Figure 6F*, *p<0.05, ordinary two-way ANOVA). IWC (LAC-4)-immunization-induced rapid TNF-α expression at 2 hpi and neutrophil infiltration at 4 hr post *A. baumannii* challenge were dismissed in *Tlr4*[-/-] mice (*Figure 6G, H*). In addition, we found that IWC-priming AMs from *Tlr4*[-/-] mice significantly lost TNF-α secretion in response to IWC restimulation ex vivo (*Figure 6I*). These results suggest that TLR4 signaling is vital for IWC-trained AMs. Taken together, these results suggest that upregulation of TLR4 expression on trained AMs plays an important role in vaccination-induced rapid protection.

## Vaccination-induced rapid protection against *P. aeruginosa* and *K. pneumoniae* infection

Further, we tested whether intranasal vaccination could induce rapid protection in other bacterial pneumonia models. We immunized mice i.n. with IWC of *P. aeruginosa* (IWC(*P.a*)), *K. pneumoniae* (IWC(*K.p*)), *S. aureus* (IWC(*S.a*)), or *S. pneumoniae* (IWC(*S.p*)) and challenged the mice with the same bacteria 7 days after immunization. Rapid and efficient protection after intranasal immunization was also observed in *P. aeruginosa* (*Figure 7A*) and *K. pneumoniae*-infected pneumonia models (*Figure 7B*). However, immunization could not induce effective protection against *S. aureus* (*Figure 7C*) and *S. pneumoniae* (*Figure 7D*). Since *A. baumannii, P. aeruginosa,* and *K. pneumoniae* are Gram-negative

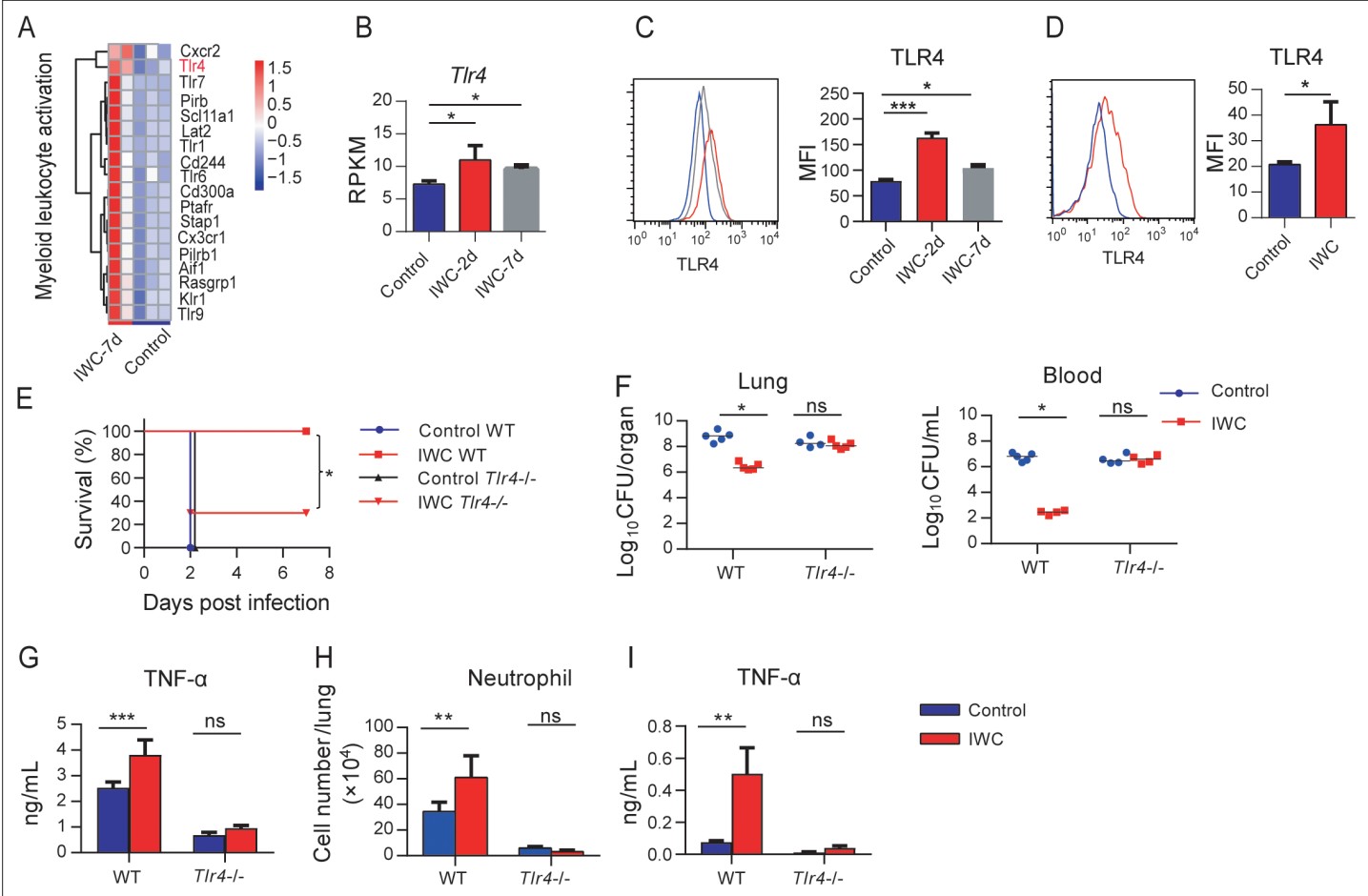

**Figure 6.** Higher TLR4 expression on inactivated whole cell (IWC)-trained alveolar macrophages (AMs) mediates rapid protection. (**A**) Heatmap of differentially expressed genes (DEGs) associated with myeloid leukocyte activation at day 7 after *A. baumannii* IWC (LAC-4) immunization in *Rag1*[-/-] mice. (**B**) Reads per kilobase per million reads (RPKM) of *Tlr4* in lungs on days 2 and 7 after IWC (LAC-4) immunization. (**C**) Representative histogram of TLR4 expression and mean fluorescence index (MFI) of TLR4 on AMs in BALF on day 2 (red line) and day 7 (gray line) after IWC (LAC-4) immunization or control (blue line). n = 3. For (**B**) and (**C**), *p<0.05, **p<0.01, evaluated by ordinary one-way ANOVA. (**D**) Representative histogram of TLR4 expression and MFI of TLR4 on AMs after IWC stimulation in vitro (red line) for 7 days. n = 3. Data are mean ± SD. *p<0.05, determined by two-tailed unpaired *t* test. (**E–H**) *Tlr4*[-/-] and WT mice were immunized intranasally (i.n.) with IWC (LAC-4) and challenged with LAC-4 7 days later. (**E**) Survival curve (n = 5 for WT mice, n = 10 for *Tlr4*[-/-] mice). (**F**) Bacterial burdens at 24 hr post infection (hpi). (**G**) TNF-α in lungs at 2 hpi. (**H**) Neutrophil infiltration in lungs at 4 hpi. n = 4–5 mice for (**F–H**). (**I**) TNF-α levels in 2 hr culture supernatants of ex vivo IWC (LAC-4)-stimulated AMs from 7 -day-vaccinated WT or *Tlr4*[-/-] mice. n = 3. For survival, p-value was calculated by log-rank test. From (**F**) to (**I**), *p<0.05, **p<0.01, ***p<0.001, ns, not significant, compared by ordinary two-way ANOVA. In (**F**), each dot represents one mice and the line means median. Data are representative of at least two independent experiments.

The online version of this article includes the following source data and figure supplement(s) for figure 6:

**Source data 1.** Raw data for *Figure 6*.

**Figure supplement 1.** Upregulated differentially expressed genes (DEGs) at day 7 after inactivated whole cell (IWC) immunization in *Rag1*[-/-] mice.

bacteria and *S. aureus* and *S. pneumoniae* are Gram-positive bacteria, we reasoned that intranasal vaccination-induced rapid protection might be an effective way to protect certain Gram-negative bacterial pneumonia.

## Cross-protection against other bacteria induced by intranasal vaccination of IWC of *A. baumannii*

An important feature of trained immunity is the nonspecific protection to other pathogens different from the previously encountered stimulus. Therefore, we tested the cross-protection of IWC of *A. baumannii* LAC-4-induced rapid protection against other bacterial pneumonia. Mice immunized with IWC of *A. baumannii* LAC-4 (IWC(*A.b*)) showed partial protection against *P. aeruginosa* (*Figure 8A*),

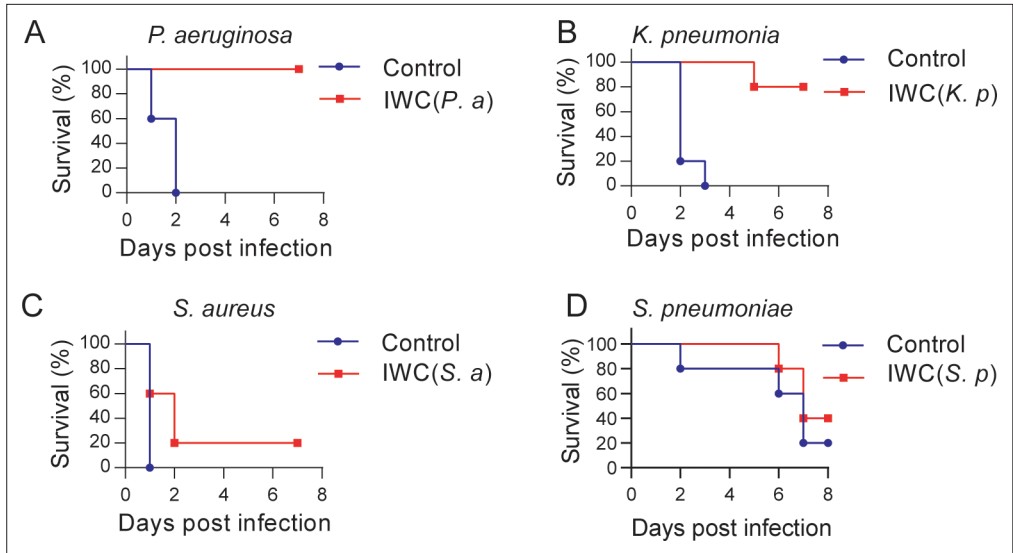

**Figure 7.** A rapid protection induced by intranasal vaccination against other bacteria. C57BL/6 mice were immunized intranasally (i.n.) with inactivated whole cell (IWC) of (**A**) *P. aeruginosa* (IWC(P.a)), n = 5, (**B**) *K. pneumoniae* (IWC(K.p)), n = 5, (**C**) *S. aureus* (IWC(S.a)), n = 10, or (**D**) *S. pneumoniae* (IWC(S.p)), n = 5. On day 7, the immunized mice were challenged intratracheally (i.t.) with the same species. The survival rates were monitored for 7 days. Data are representative of at least two independent experiments.

The online version of this article includes the following source data for figure 7:

**Source data 1.** Raw data for *Figure 7*.

*K. pneumoniae* (*Figure 8B*), *S. aureus* (*Figure 8C*), and *S. pneumoniae* (*Figure 8D*) challenge at day

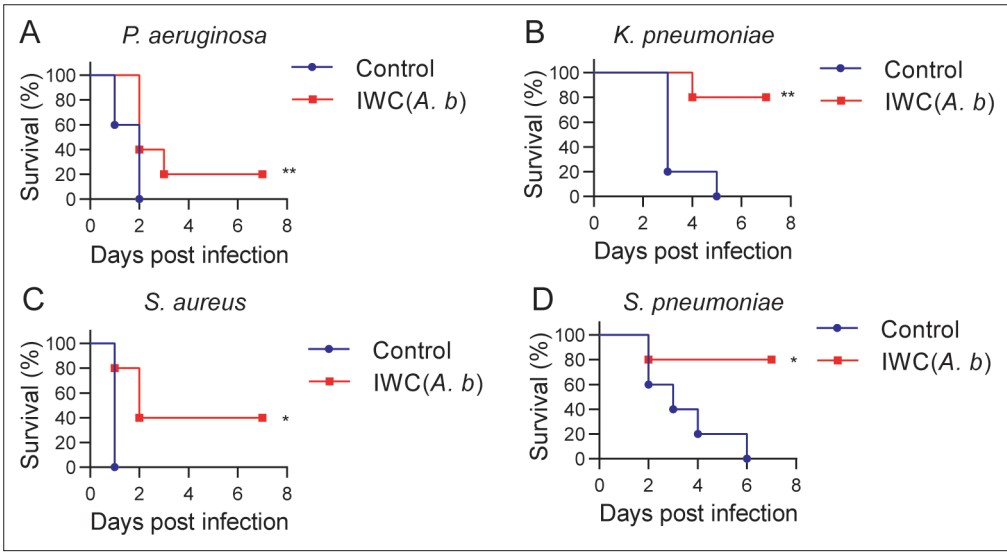

**Figure 8.** Cross-protection induced by intranasal vaccination of inactivated whole cell (IWC) of *A. baumannii*. C57BL/6 mice were immunized intranasally (i.n.) with IWC of *A. baumannii* LAC-4 (IWC(A.b)) and challenged intratracheally (i.t.) with (**A**) *P. aeruginosa* (1 × 10⁷ CFU, n = 10), (**B**) *K. pneumoniae* (1 × 10⁷ CFU, n = 5), (**C**) *S. aureus* (5 × 10⁷ CFU, n = 5), or (**D**) *S. pneumoniae* (2 × 10⁷ CFU, n = 5) at day 7 after immunization. Survival was recorded for 7 days. *p<0.05, **p<0.01, determined by log-rank test. Data are representative of at least two independent experiments.

The online version of this article includes the following source data for figure 8:

**Source data 1.** Raw data for *Figure 8*.

7 after immunization. These data indicate that IWC of *A. baumannii* vaccination also induced nonspecific protection against other bacterial infection. So the training of innate immune response with IWC of *A. baumannii* can also confer cross-protection against other bacteria except *A. baumannii*.

## Discussion

The challenge of MDR bacterial infection highlights the urgent need to develop rapid-acting vaccine. Currently, only some vector-based virus vaccines are reported to be able to elicit rapid protection by a single dose, such as Ebola and Zika virus vaccines (*Marzi et al., 2015*; *Pardi et al., 2017*; *Wong, 2019*). Here, we showed that a single intranasal immunization of IWC of *A. baumannii* elicits a very rapid and complete protection against *A. baumannii* infection 2 or 7 days after vaccination, supported by 100% survival, reduced bacterial burdens, alleviated lung injury, and reduced inflammatory cytokines expression after challenge (*Figure 1*). In addition, the IWC vaccination-induced rapid protection is broad against different clinical strains including different antibiotic-sensitive or -resistant strains.

The development of immunological memory by vaccination is a central goal in fighting against infections. *A. baumannii* IWC-induced fast protection leads us to suspect whether it is a result of sustained and activated immune response elicited by vaccination. Dynamic response to intranasal immunization of IWC of *A. baumannii* shows that host response rapidly undergoes the priming, resting, and results in memory stage 5 days later (*Figures 3A, 4F*). The innate immune response 2 days post vaccination might be an activated innate immune response reflected by increase IL-6 and TNF-α response to vaccination (*Figure 3A*), which provides the host resistance to *A. baumannii* infection. The host response to vaccination completely recovered to baseline level at day 5 post vaccination. However, IWC still induces protection against *A. baumannii* infection at day 7 after immunization, which represents a recall response to vaccination. Upon reexposure to *A. baumannii* 7 days after vaccination, IWC-vaccinated mice recalls a rapid, heightened TNF-α secretion and chemokine production at 2 hr post challenge and subsequently increased neutrophils infiltration earlier at 4 hr post challenge in lungs of vaccinated mice (*Figure 3B–D*). It is well known that neutrophil, macrophage, and monocytes play essential roles in host defense against *A. baumannii* infection (*Chen, 2020*; *Qiu et al., 2012*; *van Faassen et al., 2007*). So, vaccination-induced rapid recall responses to infection lead to a rapid elimination of bacteria, thereby limit the uncontrolled inflammation at 24 hpi in vaccinated mice, eventually prevent lung damage and efficiently improve the survival of mice.

For mechanisms study, we highlight the role of trained innate immunity in vaccination-induced rapid protection. Traditionally, immune memory is thought to be an exclusive feature of adaptive immune response of T cells and B cells, which is harnessed to design the vaccine extensively, whereas the role of innate immune response to vaccination is recognized as modulation of adaptive immunity. Recently, trained immunity has been proposed to describe the enhanced immune response of innate cells to second stimuli via epigenetic, metabolic, and functional reprogramming by initial stimulation (*Mulder et al., 2019*; *Netea et al., 2016*). Our study in *Rag1*$^{-/-}$ mice (which lack mature T and B cells) further showed innate immune responses can be trained by vaccination to mediate rapid protection (*Figure 4*). A hallmark characteristic of trained immunity is that trained immunity is nonspecific and thus theoretically effective against all types of pathogens. Our data showed that intranasal immunization with IWC of *A. baumannii* can confer broad cross-protection against other bacteria except *A. baumannii* although at a relative lower protective rate (*Figure 8*). The rapidity of innate response with the trained feature enables it as a good target to design the vaccine to induce rapid protective response against MDR bacteria.

AMs are the predominant cells in the airway mucosa and play important roles for infection controls (*Hussell and Bell, 2014*). The embryonic origin and the ability of self-renewal in steady state of AMs make them be able to store immune memory (*Hussell and Bell, 2014*). In this study, we found that AMs could be trained by IWC of *A. baumannii* with functional reprogramming, showing increased TNF-α production upon restimulation with the same IWC. More importantly, adoptive transfer of *A. baumannii* IWC-trained AMs into naive recipient mice enhanced the bacteria clearance after lethal challenge with *A. baumannii* (*Figure 5*), confirming that IWC-trained AMs mediate rapid protection. Currently, there are limited reports about trained immunity of AMs. It has been reported that AMs could be trained to gain the memory phenotype in an adenovirus-infected models, which is dependent on IFN-γ production from effector CD8$^+$ T cells (*Yao et al., 2018*). This data, together with our findings, supports the trained immunity of AMs. However, the training of AMs with IWC of bacteria

has distinct feature from virus-trained AMs. Training of AMs with IWC of bacteria in this study is independent on adaptive T cells since AMs could be trained by IWC directly in vitro (*Figure 5C*) and vaccination-induced protection is still efficient in *Rag1*$^{-/-}$ mice (*Figure 4A*). The trained feature of AMs can be manipulated to combat MDR bacterial pneumonia and also the respiratory virus infection.

As for how AMs are trained, a range of pattern recognition receptors (PRRs), including Toll-like receptors (TLRs), nucleotide-binding oligomerization domain-containing protein 2 (NOD2), and dectin-1 might be engaged to promote trained immunity. Bacille Calmette-Guérin (BCG) and its main component muramyl dipeptide induce trained immunity through NOD2-ligand (*Kleinnijenhuis et al., 2012*) and β-glucan through dectin-1 receptor (*Quintin et al., 2012*). Some studies have implicated that TLRs are upregulated to BCG or β-glucan training (*Kleinnijenhuis et al., 2012*; *Quintin et al., 2012*); however, how TLRs are involved in trained immunity is not clear. In this study, we demonstrate that TLR4 is elevated on IWC-trained AMs. Enhanced TNF-α production of vaccine-primed AMs upon restimulation is impaired in *Tlr4*$^{-/-}$ mice, which results in reduced protective effect of vaccination (*Figure 6*). These data suggest that elevated TLR4 expression on AMs might be a trained marker, which can sense the pathogen-associated molecular pattern more efficiently and results in rapid activation of trained AMs compared to naïve AMs. TNF-α is critical for an effective antibacterial host defense against bacterial infection by regulating downstream cytokine response and activating immune cells. TNF-α is one of the main cytokines that has been thoroughly used as a functional cytokine marker indicating trained immunity along with IL-6 and IL-1β (*Arts et al., 2018*). In this study, we found that vaccination-trained immune response produces heightened TNF-α, but not IL-6 and IL-1β, when encountered with the infection (*Figure 3B*). In addition, blocking TNF-α with specific antibody before challenge significantly reduces vaccine-induced rapid protection (*Figure 5F*). These data collectively indicate that enhanced production of TNF-α from AMs is a functional indicator of trained AMs and is responsible for vaccine-elicited rapid protection.

What's more important in this study, we found that the vaccination-induced rapid protection is also observed in *P. aeruginosa* and *K. pneumoniae*-infected pneumonia models but not in *S. aureus* and *S. pneumoniae*-infected pneumonia models (*Figure 7*). Since limited data of how AMs were trained are reported, we do not know why some Gram-positive bacteria cannot train AMs to exert rapid protection. We guess that the different cell wall component in Gram-negative and Gram-positive bacteria may account for the different vaccination effect of different bacteria. In addition, it might involve the different host response to different infection. Some cytokines like human IL-37 have been reported as an inhibitor of trained immunity (*Cavalli et al., 2021*). Further studies are needed to explore the different host response to different vaccination.

The limitation of this study is that we could not test the vaccination-induced rapid protection in more MDR bacterial pneumonia models due to the lack of more bacterial pneumonia models in our hands. Also, our study also leaves open many questions, such as how AMs are trained by vaccination, what ligand-receptor pairs are responsible for training AM, and what molecular mechanism is involved. Further investigation is needed to better understand the trained immunity of AMs, which in turn will pave the way for improved vaccine design.

In summary, in this study we demonstrate that intranasal immunization of IWC of certain bacteria induces a rapid, sufficient, and broad protection against lethal respiratory infection through inducing trained immunity of AMs. Our study highlights the importance and the possibility of harnessing trained immunity of AMs to design rapid-effecting vaccine against MDR bacterial pneumonia for inpatients. Even for long-lasting effect of vaccine, exploiting the classical adaptive memory and trained innate immunity in an integrated fashion seems plausible for a potential good design of vaccination strategies against bacterial infection.

## Materials and methods
### Experimental design and ethical approval

This study was designed to determine whether intranasal immunization of IWC could elicit rapid protection against bacterial pneumonia and explore the underlying mechanisms. For animal studies, 6–8- week-old, female C57BL/6, *Rag1*$^{-/-}$, and *Tlr4*$^{-/-}$mice were used. The animal protocols adhered to the National Institutes of Health Guide for the Care and Use of Laboratory Animals and were approved by the Institutional Animal Care and Treatment Committee of West China Hospital, Sichuan

University (approval no. 2019190A). An MS Excel randomization tool was used to randomize the mice to different treatment groups. To assess the protection efficacy of IWC, survival rate, clinical score, bacterial burdens, and lung histopathology of mice were monitored after infection. RNA-seq, ex vivo, and in vitro AMs stimulation model and adoptive transfer of AMs were performed to investigate the mechanisms underlying the rapid protection. The animal studies were not blinded. The group sizes for survival varied from 5 to 10 in the different studies and 3–5 for analysis of immune response. All experiments were conducted at least two times independently, which is indicated in the figure legends.

## Mice

Female C57BL/6 mice were purchased from Beijing HFK Bioscience Limited Company (Beijing, China). *Rag1* gene knockout mice (*Rag1*[-/-], B6.129S7-*Rag1*[tm1]/Nju), TLR4 gene knockout mice (*Tlr4*[-/-], C57BL/10ScNJNju), and control WT mice were purchased from the Model Animal Research Center of Nanjing University. The mice were kept under specific pathogen-free conditions.

## Bacterial strains

*A. baumannii* strain LAC-4 was kindly provided by Professor Chen (*Harris et al., 2013*). Reference strain of *A. baumannii* ATCC17978 was purchased from ATCC. Clinical strains of *A. baumannii* SJZ24, SJZ26, and SJZ09 were isolated from Bethune International Peace Hospital in Shijiazhuang, China, in 2016 (*Zeng et al., 2019*). *P. aeruginosa* strain XN-1 (*Gao et al., 2020*), *K. pneumoniae* strain YBQ (*Liu et al., 2020*), and *S. pneumoniae* were isolated from Chongqing Southwest Hospital. *S. aureus* strain KM-22 was isolated from the Second Affiliated Hospital of Kunming Medical University (*Zeng et al., 2020*). The bacteria were grown in tryptone soy broth (*A. baumannii*), Luria-Bertani broth (*P. aeruginosa* and *K. pneumoniae*), Mueller-Hinton broth (*S. aureus*), or blood agar plate (*S. pneumoniae*) at 37°C. At mid-log-phase, bacteria were collected and suspended in phosphate buffer saline (PBS). Fresh bacteria were used to infect the mice. For IWCs preparation, fresh bacteria were fixed with 4% paraformaldehyde and washed with PBS.

## Intranasal immunization and pneumonia model

Mice were anesthetized by intraperitoneal injection of pentobarbital sodium (62.5 mg/kg of body weight) and then immunized i.n. with IWC ($1 \times 10^8$ CFUs in 20 µl PBS) or PBS as a control. Mice were infected with a lethal dose of bacteria intratracheally through mouth via a soft-end needle under direct visualization to establish pneumonia model (*Gu et al., 2018*). The survival rate, clinical score, bacteria burdens, and lung pathology were evaluated as described previously (*Gu et al., 2018*). The lethal doses for different bacteria are as follows: $2 \times 10^7$ CFUs for *A. baumannii*; $1 \times 10^7$ CFUs for *P. aeruginosa*; $1 \times 10^7$ CFUs for *K. pneumoniae*; $5 \times 10^7$ CFUs for *S. aureus*; or $2 \times 10^7$ CFUs for *S. pneumoniae*.

## ELISA

TNF-α, IL-6, and IL-1β concentrations in serum, lung homogenates, and cell culture supernatants were detected using mouse TNF-α ELISA kit, mouse IL-6 ELISA kit, and mouse IL-1β ELISA kit (eBioscience, San Diego, CA) following the manufacturer's instructions.

## Real-time PCR

Total RNA of lungs was extracted by RNAiso Plus (Takara Biotechnology, Dalian, China) and reverse transcribed to cDNA with PrimeScript RT reagent Kit (Takara Biotechnology). Gene expression was detected using TB Green Premix Ex Taq II (Takara Biotechnology) on CFX96 real-time PCR detection machine (Bio-Rad, Hercules, CA) with specific primers listed in *Supplementary file 1*. The ΔΔCt method was used to calculate the relative gene expression with β-actin as the housekeep gene.

## Preparation of BALF and lung cell suspension

Cells were obtained from BALF as described (*Gu et al., 2018*). Perfused lungs were cut into small pieces and digested with 1 mg/mL collagenase D (Sigma-Aldrich, St. Louis, MO) and 100 µg/mL DNAase (Sigma) at 37°C for 60 min. Cell suspension was prepared by crushing and filtering the digested tissue through a 70 µm cell strainer (BD Biosciences, NJ) and the cell numbers were counted by Countess II Automated Cell Counter (Thermo Fisher Scientific, MA).

## Flow cytometry

Cell suspensions were blocked with rat serum then stained with fluorophore-conjugated specific or isotype control antibodies in the dark at 4 °C for 30 min. The antibodies were as follows: CD45-PE/Cy7 (30-F11), CD11b-PerCP/Cy5.5 (M1/70), F4/80-APC (BM8), Ly-6C-PE (HK1.4), Ly6G-FITC (1A8), and CD11c-PE (N418) from BioLegend (San Diego, CA) and TLR4-PE (UT41) from Invitrogen (Carlsbad, CA). Labeled cells were run on a BD FACSCanto II flow cytometer (BD Biosciences) and analyzed with FlowJo (BD Biosciences). AMs were defined as CD45$^+$CD11b$^-$F4/80$^+$; monocytes were defined as CD45$^+$CD11b$^+$ Ly6C$^+$; neutrophils were defined as CD45$^+$CD11b$^+$Ly6G$^{hi}$. The cell numbers of each cell types were calculated with total cell number multiplied by the cell percentage. AMs in BALF were identified as CD11c$^+$ F4/80$^+$ cells, and TLR4 expression on AMs were detected by flow cytometry and expressed as mean fluorescence index (MFI).

## RNA sequencing (RNA-seq)

Tissue samples from lungs were sent to Wuhan Seqhealth Co., Ltd. (Wuhan, China) for RNA-seq. Briefly, Total RNAs were extracted from lung samples using TRIzol (Invitrogen) and DNA was digested by DNaseI after RNA extraction. A260/A280 was examined with NanoDrop One spectrophotometer (Thermo Fisher Scientific) to determine RNA quality. RNA integrity was confirmed by 1.5% agarose gel electrophoresis. Qualified RNAs were finally quantified by Qubit3.0 with Qubit RNA Broad Range Assay kit (Life Technologies, Carlsbad, CA). Total RNAs (2 μg) were used for to prepare sequencing library using KC-Total RNA-seq Library Prep Kit for Illumina (Wuhan Seqhealth Co., Ltd., Wuhan, China) following the manufacturer's instruction. PCR products corresponding to 200–500 bps were enriched, quantified, and finally sequenced on HiSeq X 10 sequencer (Illumina).

## Analysis of RNA-seq data

Raw data of sequencing were cleaned using Trimmomatic software. The clean reads after quality control were mapped to the mouse genome GRCm38 with STAR software (version 2.5.3a). The reads counts for each gene were calculated using FeatureCounts (version 1.5.1) and expressed as reads per kilobase per million reads (RPKM). EdgeR package was used to identify the DEGs by statistics with an adjusted p-value<0.05 and fold change > 1.5. GO and Kyoto Encyclopedia of Genes and Genomes (KEGG) enrichment was done with Kobas (version 2.1.1) or Enrichr (https://amp.pharm.mssm.edu/Enrichr/; *Chen et al., 2013*; *Kuleshov et al., 2016*). Hierarchical clustering and heatmaps were drawing using pheatmap R package, and MA plot was drawn using EdgeR package.

## In vitro AMs training model

BALF cells were cultured in DMEM (containing 10% FBS and 1% penicillin/streptomycin) in plate for 1 hr, and nonadherent cells were discarded and remaining AMs were stimulated with LAC-4 (multiplicity of infection [MOI] = 1) for 24 hr, then washed out and rested for 6 days. AMs were restimulated with IWC (LAC-4, MOI = 1) at day 7, and TNF-α in supernatants at 2 hr after restimulation was detected by ELISA. In some experiments, AMs were collected at day 7 to detect TLR4 expression with PE-anti TLR4 antibody (UT41, Invitrogen) by flow cytometry. The expression of TLR4 is showed as MFI.

## Magnetic-activated cell sorting (MACS)

AMs from BALF were sorted by positive selection with an anti-mouse CD11c Microbeads kit (Miltenyi Biotech, Bergisch Gladbach, Germany) according to the manufacturer's instruction. Sorted cells were stained with PE-anti-mouse CD11c antibody (N418) and F4/80-APC (BM8) and analyzed by flow cytometry to check the purity.

## Stimulation of AMs ex vivo

Mice were immunized i.n. with IWC (LAC-4) or PBS as control. AMs (CD11c$^+$) sorted by MACS from BALF at day 7 were stimulated ex vivo with same IWC (MOI = 1) for 2 hr and supernatant was collected for measuring TNF-α using ELISA.

## Adoptive transfer of AMs

AMs (CD11c$^+$) from BALF of IWC (LAC-4)-immunized or control mice at day 7 after immunization were sorted by MACS as described above (purity >95%). Donor AMs (5 × 10$^4$) were i.t. transferred into the

airways of recipient mice. Recipient mice were challenged with LAC-4 ($2 \times 10^7$) at 4 hr after transfer, and bacterial burdens in lungs and clinical score were detected 24 hpi.

## Blocking TNF-α in vivo

For neutralizing TNF-α, mice were treated with 200 µg anti-mouse TNF-α (XT3.11 clone, Bio X Cell, West Lebanon, NH) or rat IgG1 isotype antibody intraperitoneally 1 hr before infection, and the survival was recorded for 7 days.

## Statistical analyses

Bacterial burdens and clinical score data were expressed as median. Other bar graph data were presented as means ± SD. Survival data were compared by log-rank test. For data of more than two groups, data were evaluated by ordinary one-way ANOVA followed by Tukey's multiple comparisons test. Data of two samples with normal distribution were compared by two-tailed unpaired $t$ test. Mann–Whitney U test was used for comparing data of non-normal distribution (bacterial burdens and clinical score). For grouped data, statistical significance was evaluated by ordinary two-way ANOVA. The software GraphPad Prism (v 8.3.0) was used for all statistical analyses. All comparisons used a two-sided α of 0.05 for significance testing, and p<0.05 was considered significant. The specific statistical methods are indicated in the figure legends.

## Acknowledgements

We thank Prof. Wangxue Chan for kindly providing the *A. baumannii* strain LAC4 and Dr. Georgina T Salazar for editing the manuscript.

## Additional information

### Funding

| Funder | Grant reference number | Author |
| --- | --- | --- |
| National Natural Science Foundation of China | 81971561 | Yun Shi |

The funders had no role in study design, data collection and interpretation, or the decision to submit the work for publication.

### Author contributions

Hao Gu, Data curation, Formal analysis, Investigation, Writing – original draft; Xi Zeng, Data curation, Formal analysis, Investigation; Liusheng Peng, Chuanying Xiang, Yangyang Zhou, Xiaomin Zhang, Jixin Zhang, Investigation, Methodology; Ning Wang, Gang Guo, Yan Li, Kaiyun Liu, Jiang Gu, Hao Zeng, Haibo Li, Jinyong Zhang, Weijun Zhang, Methodology, Resources; Yuan Zhuang, Writing – review and editing; Quanming Zou, Conceptualization, Resources, Supervision, Writing – review and editing; Yun Shi, Conceptualization, Formal analysis, Funding acquisition, Project administration, Resources, Supervision, Writing – review and editing

### Author ORCIDs

Yun Shi http://orcid.org/0000-0003-1925-4337

### Ethics

The animal protocols adhered to the National Institutes of Health Guide for the Care and Use of Laboratory Animals and were approved by the Institutional Animal Care and Treatment Committee of West China Hospital, Sichuan University (Approval No. 2019190A).

### Decision letter and Author response

Decision letter https://doi.org/10.7554/eLife.69951.sa1
Author response https://doi.org/10.7554/eLife.69951.sa2

## Additional files

### Supplementary files
- Transparent reporting form
- Supplementary file 1. Primers used in real-time PCR.

### Data availability
Raw data files for RNAseq have been deposited in the NCBI Gene Expression Omnibus under accession number GEO: GSE141729.

The following dataset was generated:

| Author(s) | Year | Dataset title | Dataset URL | Database and Identifier |
|---|---|---|---|---|
| Gu H, Shi Y | 2020 | Transcriptome response of lung in response to vaccination of inactivated whole cells of A. baumannii | https://www.ncbi.nlm.nih.gov/geo/query/acc.cgi?acc=GSE141729 | NCBI Gene Expression Omnibus, GSE141729 |

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

# Appendix 1

**Appendix 1—key resources table**

| Reagent type (species) or resource | Designation | Source or reference | Identifiers | Additional information |
|---|---|---|---|---|
| Genetic reagent (*Mus musculus*) | C57BL/6J | Beijing HFK Bioscience Limited Company | N/A | |
| Genetic reagent (*Mus musculus*) | *Rag1*-/-: B6.129S7-Rag1tm1/Nju | Model Animal Research Center of Nanjing University | Stock no :N000088 | |
| Genetic reagent (*Mus musculus*) | *Tlr4*-/- mice: C57BL/10ScNJNju | Model Animal Research Center of Nanjing University | Stock no: N000192 | |
| Genetic reagent (*Mus musculus*) | C57BL/10JNju | Model Animal Research Center of Nanjing University | Stock no: N000023 | |
| Strain, strain background (*Acinetobacter baumannii*) | LAC-4 | Gift from Professor Chen, *Harris et al., 2013* | N/A | |
| Strain, strain background (*Acinetobacter baumannii*) | ATCC-17978 | ATCC | ATCC-17978 | |
| Strain, strain background (*Acinetobacter baumannii*) | SJZ-09 | Clinical isolate, *Zeng et al., 2019* | | |
| Strain, strain background (*Acinetobacter baumannii*) | SJZ-24 | Clinical isolate, *Zeng et al., 2019* | | |
| Strain, strain background (*Acinetobacter baumannii*) | SJZ-26 | Clinical isolate, *Zeng et al., 2019* | | |
| Strain, strain background (*Pseudomonas aeruginosa*) | XN-1 | Clinical isolate, *Gao et al., 2020* | | |
| Strain, strain background (*Klebsiella pneumoniae*) | YBQ | Clinical isolate, *Liu et al., 2020* | | |
| Strain, strain background (*Staphylococcus aureus*) | KM-22 | Clinical isolate, *Zeng et al., 2020* | | |
| Strain, strain background (*Streptococcus pneumoniae*) | *S. pneumoniae* | Clinical isolate | | |
| Antibody | Rat monoclonal anti-mouse CD45-PE/Cy7 (clone 30-F11) | BioLegend | Cat# 103114 RRID:AB_312979 | (1:100) |
| Antibody | Rat monoclonal anti-mouse/ human CD11b-PerCP/Cy5.5 (clone M1/70) | BioLegend | Cat# 101228 RRID:AB_893232 | (1:100) |
| Antibody | Rat monoclonal anti-mouse F4/80-APC (clone BM8) | BioLegend | Cat# 123116 RRID:AB_893481 | (1:100) |
| Antibody | Rat monoclonal anti-mouse Ly-6C-PE (clone HK1.4) | BioLegend | Cat# 128008 RRID:AB_1186132 | (1:100) |
| Antibody | Rat monoclonal anti-mouse Ly-6G-FITC (clone 1A8) | BioLegend | Cat# 127605 RRID:AB_1236488 | (1:100) |
| Antibody | Hamster monoclonal anti-mouse CD11c-PE (clone N418) | BioLegend | Cat# 117307 RRID:AB_313776 | (1:100) |
| Antibody | Rat monoclonal anti-mouse CD11c-FITC (clone N418) | BioLegend | Cat# 117305 RRID:AB_313774 | (1:100) |
| Antibody | Mouse monoclonal anti-mouse CD284(TLR4)-PE (clone UT41) | Thermo Fisher | Cat# 12-9041-80 RRID:AB_466236 | (1:100) |
| Antibody | Rat monoclonal rat IgG1 isotype control (clone HRPN) | Bio X Cell | Cat# BE0088, RRID:AB_1107775 | 200 µg/per mouse |

*Appendix 1 Continued on next page*

*Appendix 1 Continued*

| Reagent type (species) or resource | Designation | Source or reference | Identifiers | Additional information |
|---|---|---|---|---|
| Antibody | Rat monoclonal anti-mouse TNF-$\alpha$ (clone XT3.11) | Bio X Cell | Cat# BE0058, RRID:AB_1107764 | 200 µg/per mouse |
| Sequence-based reagent | Primers for qPCR | This paper | N/A | See *Supplementary file 1* |
| Commercial assay or kit | RNAiso Plus | Takara | Cat# 9109 | |
| Commercial assay or kit | PrimeScript RT reagent kit | Takara | Cat# RR037A | |
| Commercial assay or kit | TB Green Premix Ex Taq II | Takara | Cat# RR820A | |
| Commercial assay or kit | IL-6 ELISA kit | BioLegend | Cat# 431304 | |
| Commercial assay or kit | TNF-$\alpha$ ELISA kit | BioLegend | Cat# 430904 | |
| Commercial assay or kit | IL-1β ELISA kit | BioLegend | Cat# 432604 | |
| Commercial assay or kit | Anti-mouse CD11c Microbeads kit | Miltenyi Biotec | Cat# 130-125-835 | |
| Software, algorithm | GraphPad Prism | GraphPad Software | RRID:SCR_002798 | |
| software, algorithm | FlowJo | FlowJo | RRID:SCR_008520 | |

