## [Decision Letter]

**Acceptance summary:**

The development and spread of resistance against most antibiotics makes the management of bacterial infections a challenging issue. Here, Gu et al. describe the use of intranasal immunization of IWC of *A. baumannii*, *P. aeruginosa* and *K. pnemoniae* against infection by *A. baumannii*, *P. aeruginosa* and *K. pnemoniae*, respectively. These findings build on our understanding of immunity to bacterial pathogens and may be used to contribute to future vaccine strategies.

**Decision letter after peer review:**

Thank you for submitting your article "Vaccination-induced rapid protection against bacterial pneumonia via training alveolar macrophage in mice" for consideration by *eLife*. Your article has been reviewed by 2 peer reviewers, and the evaluation has been overseen by a Reviewing Editor and Tadatsugu Taniguchi as the Senior Editor. The reviewers have opted to remain anonymous.

Essential revisions:

1. More clarity is required to determine how numbers of animals were chosen and whether the studies are powered to determine significant differences. Repeats of experiments should be performed and noted to allow for evaluation of reproducibility.

2. Additional experiments involving cross-protection (non-specific protection) are required to evaluate the role for trained innate immunity. As part of this, the authors should discuss why immunization with IWC of *S. aureus* and *S. pneumoniae* did not induce effective protection against *S. aureus* and *S. pneumoniae*.

3. It has been reported that memory alveolar macrophages required T cell help for trained immunity, the authors should expand on the mechanism in their work that makes it distinct from these previous findings.

Although the article is well presented, there are a number of pieces of information that are required to support the authors' interpretations and conclusions. Specifically:

1. The authors should compare the transcriptional profiles/inflammatory responses and histopathology of vaccinated wt and Rag1^-/-^ mice.

2. Figure 4E. The authors should determine the survival of mice with or without i.t. AM transfer and challenged by *A. baumannii*.

3. The authors need to discuss why immunization with IWC of *S. aureus* and *S. pneumoniae* did not induce effective protection against *S. aureus* and *S. pneumoniae*. Could the difference in the membrane structure of Gram-negative and Gram-positive bacteria be the responsible in the observed effect of the immunization with IWCs?

4. In order to support a broader scope of protection, the authors could perform an additional experiment to determine whether the immunization with IWC of *A. baumannii* LAC-4 strain might protect against infections by other MDR clinical isolates of *A. baumannii*.

5. The authors should provide the rationale for challenging the vaccinated animal on day 1 and 7

6. More clarity is required to determine how numbers of animals were chosen and whether the studies are powered to determine significant differences. Repeats of experiments should be performed and noted to allow for evaluation of reproducibility.

7. Additional experiments involving cross-protection (non-specific protection) are required to evaluate the role for trained innate immunity.

8. It has been reported that memory alveolar macrophages required T cell help for trained immunity, the authors should expand on the mechanism in their work that makes it distinct from these previous findings.

9. In SFigure 1. IL-6, TNF-α, and IL1-β are analyzed, the authors should explain how they concluded that TNF-α was specifically responsible for protection?

10. In the Blocking TNF-α in vivo study, the authors should show survival and histopathology results.

11. It has been reported that IL-37 is an inhibitor of trained immunity, IL-37 could be measured in the Gram positive experiments to investigate why protection does not occur.

---

## [Author Response]

Essential revisions:1. More clarity is required to determine how numbers of animals were chosen and whether the studies are powered to determine significant differences. Repeats of experiments should be performed and noted to allow for evaluation of reproducibility.

For the IWC of *A. baumannii*-induced protection assay, at beginning, we usually used 10 mice per group for survival evaluation. Since the protection of IWC of *A. baumannii*-immunization is 100% and repeatable. Then for the further study of IWC immunization in WT mice, we usually use 5 mice per group to save mice and also get clear results since the protection is almost always 100%. For *Rag1^-/-^*,*Tlr4^-/-^* ,or anti-TNF in vivo blocking experiment, we are not sure whether there is any influence of these factors on protection and we usually used larger numbers to make sure that the study is powered. We have clarified the numbers of animals of each experiment in the figure legends in revised manuscript.

2. Additional experiments involving cross-protection (non-specific protection) are required to evaluate the role for trained innate immunity. As part of this, the authors should discuss why immunization with IWC of *S. aureus* and *S. pneumoniae* did not induce effective protection against *S. aureus* and *S. pneumoniae*.

Thank you for the constructive suggestion. We already did these experiments. The immunization of IWC of *A. baumannii* also induces non-specific protection against other bacterial infection at a lower protection rate compared to protection against *A. baumannii.* We included these data in this version of manuscript as new figure 8.

As for why immunization with IWC of *S. aureus* and *S. pneumoniae* did not induce effective protection against *S. aureus* and *S. pneumoniae*, we reasoned that different cell wall component of Gram-negative and Gram-positive bacteria might account for the different effect of the immunization with IWC. In addition, different host response to these bacteria might also contribute to the different effects. We have added more discussion in the Discussion section in revised manuscript.

3. It has been reported that memory alveolar macrophages required T cell help for trained immunity, the authors should expand on the mechanism in their work that makes it distinct from these previous findings.

For the mechanisms study, we found that IWC immunization can induce protection against *A. baumannii* infection in *Rag1*^-/-^ mice, which lack of adaptive immune response of T cell and B cells. This result indicates that IWC of bacteria-induced trained immunity is independent on T cell. We also found that IWC can train alveolar macrophages in vitro directly. In the paper of Yao et al., they found that the training of AMs by an adenovirus requires adaptive response of CD8^+^ T cells. So the mechanism of AMs training is different. In addition, in Yao`s study, elevated MHC-II expression is a marker of trained immunity. In this study, we focused on the pattern recognition receptors and found elevated TLR4 expression is induced by IWC-immunization and which is responsible for the protection. So in our model of vaccination-induced alveolar macrophages training, there are totally different mechanisms involved. Also, there must be other distinct mechanisms needs to be further studied. We have added more discussion of these in the discussion part of revised manuscript.

Although the article is well presented, there are a number of pieces of information that are required to support the authors' interpretations and conclusions. Specifically:1. The authors should compare the transcriptional profiles/inflammatory responses and histopathology of vaccinated wt and Rag1^-/-^ mice.

Thank you for the suggestion. In this study, we did IWC immunization in *Rag1*^-/-^ mice to find out whether adaptive immune response or innate immune response is responsible for the rapid protection against *A. baumannii* infection. The results showed that IWC also induced good protection in *Rag1*^-/-^ mice, indicating that adaptive immune response is not required for the rapid protection. So in the further study, we explored the transcriptional profile of host response to immunization and infection in *Rag1*^-/-^ mice with RNAseq to exclude the effect of adaptive immune response. We did not do the same transcriptional profile in WT mice, since we think that comparing the difference of transcriptional profile of vaccinated WT and *Rag1*^-/-^ mice is to find out the effect of adaptive immune response, which is not the purpose of this study. The RNAseq results from *Rag1*^-/-^ mice showed genes associated with inflammatory response (including *Il6*, *Cxcl1*, *Cxcl2*,*Cxcl10*, *Ccl2*) were significantly downregulated in vaccinated mice at 24 hour post infection (Figure 4D and E), which is consistent with what we observed in WT mice detected by real-time PCR (Figure 1), indicating that the vaccination induced similar protection in WT and *Rag1*^-/-^ mice.

2. Figure 4E. The authors should determine the survival of mice with or without i.t. AM transfer and challenged by A. baumannii.

We observed the survival of the mice transferred with or without i.t. AMs and challenged by *A. baumannii* as the experiment condition in Figure 4E and did not find the difference of survival between two groups. But the difference of bacterial burdens is much easier to be observed. So we used the bacterial burdens as readout.

3. The authors need to discuss why immunization with IWC of *S. aureus* and *S. pneumoniae* did not induce effective protection against *S. aureus* and *S. pneumoniae*. Could the difference in the membrane structure of Gram-negative and Gram-positive bacteria be the responsible in the observed effect of the immunization with IWCs?

Sure, we agree with reviewer`s opinion. We reasoned that different cell wall component of Gram-negative and Gram-positive bacteria might account for the different effect of the immunization with IWC. In addition, different host response to these bacteria might also contribute to the different effects. We have added more discussion in the Discussion section in revised manuscript.

4. In order to support a broader scope of protection, the authors could perform an additional experiment to determine whether the immunization with IWC of A. baumannii LAC-4 strain might protect against infections by other MDR clinical isolates of A. baumannii.

Thank you for the suggestion. We also care about the broad protection of IWC immunization. We have the data about protection of immunization of IWC of *A. baumannii* strain of LAC-4 to some clinical isolates. Due to the less virulence of clinical isolates, we also immunized mice with IWC of different *A. baumanni* clinical isolates and test their protection to LAC-4 infection. Clinical isolate SJZ24 is an extremely antibiotic resistant clinical isolate (Zeng et al., 2019), LAC-4 is a multidrug resistant stain, and ATCC17978 is a reference strain which is an antibiotic sensitive strain (Harris et al., 2013). Together, these data suggested that intranasal immunization of IWC of *A. baumannii* can elicit broad protection against different clinical isolates independent of drug resistance. We added these data into the revised manuscript as Figure 2.

5. The authors should provide the rationale for challenging the vaccinated animal on day 1 and 7

In this study, we focused the rapid effect of vaccination against bacterial infection. At the beginning of this project, we just try to reduce the immunization does from 3 doses to 2 does or 1 dose to evaluate the effect of vaccination. We found that intranasal immunization of neither 3 doses, 2 doses, or 1 dose all elicit 100% protection against *A. baumannii* infection evaluated 7 days after last immunization. So we focused on the single immunization, then we tried to shorten the days of evaluation to detect how fast the vaccination can exert protective effect. Further we evaluated the effect on day 2, day 1 after immunization of IWC and also found the protective effect against *A. baumannii* challenge. We added these rationales in the revised manuscript.

6. More clarity is required to determine how numbers of animals were chosen and whether the studies are powered to determine significant differences. Repeats of experiments should be performed and noted to allow for evaluation of reproducibility.

For IWC of *A. baumannii*-induced protection assay, at beginning, we usually used 10 mice per group for survival evaluation. Since the protection of IWC of *A.baumannii*-immunization is 100% and repeatable. Then for the further study of IWC immunization in WT mice, we usually use 5 mice per group to save mice and also get clear results since the protection is almost always 100%. For *Rag1^-/-^*,*Tlr4^-/-^* ,or anti-TNF in vivo blocking experiment, we are not sure whether there is any influence of these factors on protection and we usually used larger numbers to make sure that the study is powered. We have clarified the numbers of animals of each experiment in the figure legends in revised manuscript.

7. Additional experiments involving cross-protection (non-specific protection) are required to evaluate the role for trained innate immunity.

Thank you for the constructive suggestion. We already did these experiments. The immunization of IWC of *A. baumannii* also induces non-specific protection against other bacterial infection at a lower protection rate compared to protection against *A. baumannii.* We included these data in this version of manuscript as new figure 8.

8. It has been reported that memory alveolar macrophages required T cell help for trained immunity, the authors should expand on the mechanism in their work that makes it distinct from these previous findings.

For the mechanisms study, we found that IWC immunization can induce protection against *A. baumannii* infection in *Rag1*^-/-^ mice, which lack of adaptive immune response of T cell and B cells. This result indicates that IWC of bacteria-induced trained immunity is independent on T cell. We also found that IWC can train alveolar macrophages in vitro directly. In the paper of Yao et al., they found that the training of AMs by an adenovirus requires adaptive response of CD8^+^ T cells. So the mechanism of AMs training is different. In addition, in Yao`s study, elevated MHC-II expression is a marker of trained immunity. In this study, we focused on the pattern recognition receptors and found elevated TLR4 expression is induced by IWC-immunization and which is responsible for the protection. So in our model of vaccination-induced alveolar macrophages training, there are totally different mechanisms involved. Also, there must be other distinct mechanisms needs to be further studied. We have added more discussion of these in the discussion part of revised manuscript.

9. In SFigure 1. IL-6, TNF-α, and IL1-β are analyzed, the authors should explain how they concluded that TNF-α was specifically responsible for protection?

In SFigure 1. we measured the inflammatory cytokine of IL-6, TNF-α, and IL1-β in IWC-immunized or control mice at 24 hr post infection, and found that IL-6 in lungs and blood were both significant reduced in IWC-immunized group compared to control group. Whereas TNF-α in blood showed significant reduction, but there is no big difference in lungs. These data suggest IWC immunization can reduce bacterial infection-induced inflammatory response consistent with reduced bacterial burdens and alleviated lung injury. As here IL-6, TNF-α, and IL-1β were used as the indicators of inflammatory cytokine to reflect inflammation induced by bacterial infection. As for the cytokines responsible for protection, we reasoned that vaccination-induced earlier and stronger response to infection can combat the bacteria at early stage of infection. So we detected the early stage of cytokine response (2 hr post infection) and found that IWC-immunized mice has enhanced TNF-α, but not IL-6 and IL-1β release in response to infection (Figure 3B). In addition, we isolated alveolar macrophages from IWC-immunized mice and stimulated ex vivo with IWC and found elevated TNF response, but not IL-6 and IL-1β response (Figure 5). Further blocking TNF-α effect in IWC-immunized mice reduced IWC-induced protection. All these data lead us to make a conclusion that IWC-immunization induced rapid TNFα response is responsible for the rapid protection against infection. We have added some discussion of these in revised manuscript in results and Discussion section.

10. In the Blocking TNF-α in vivo study, the authors should show survival and histopathology results.

We measured the survival and showed survival data in Figure 5F.

11. It has been reported that IL-37 is an inhibitor of trained immunity, IL-37 could be measured in the Gram positive experiments to investigate why protection does not occur.

We thank the reviewers for the suggestion. This is another good question to illustrate the mechanisms of why there is a big difference of trained immunity induced by Gram-positive versus Gram-negative bacteria. There might be many underlining mechanisms such as different components of cell wall in Gram positive and Gram negative bacteria. Also, it might involve the different host response to infection, such as different anti-inflammatory cytokine responses like IL-37. We have did reference research that there is no homolog gene for human IL-37 in the mouse and it was thus not possible to directly measure IL-37 response in our mouse model to Gram positive bacterial infection (Cavalli et al., 2021; Nold et al., 2010; Cavalli, & Dinarello, 2018).This is constructive suggestion for ongoing study to detect whether the different cytokine response to different bacteria might be responsible for the different protective effects. We make a little bit discussion of this question in Discussion section and will test the role of IL-37 on IWC-induced protection using supplement with human IL-37 or human IL-37 transgenic mice in future.

References

1. Zeng, X.*, et al.* A lethal pneumonia model of *Acinetobacter baumannii*: an investigation in immunocompetent mice. *Clin Microbiol Infect*
**25**, 516 e511-516 e514 (2019).

2. Harris, G.*, et al.* A mouse model of *Acinetobacter baumannii*-associated pneumonia using a clinically isolated hypervirulent strain. *Antimicrob Agents Chemother*
**57**, 3601-3613 (2013).

3. Cavalli, G.*, et al.* The anti-inflammatory cytokine interleukin-37 is an inhibitor of trained immunity. *Cell reports*
**35**, 108955 (2021).

4. Nold, M.F.*, et al.* IL-37 is a fundamental inhibitor of innate immunity. *Nat Immunol*
**11**, 1014-1022 (2010).

5. Cavalli, G. & Dinarello, C.A. Suppression of inflammation and acquired immunity by IL-37. *Immunol Rev*
**281**, 179-190 (2018).